# HPV Vaccination Behavior, Vaccine Preference, and Health Beliefs in Chinese Female Health Care Workers: A Nationwide Cross-Sectional Study

**DOI:** 10.3390/vaccines11081367

**Published:** 2023-08-15

**Authors:** Xiaoping Shao, Xinyue Lu, Weiyu Zhou, Weifeng Huang, Yihan Lu

**Affiliations:** 1Department of Intensive Care Medicine, The Sixth People’s Hospital, Shanghai Jiao Tong University, Shanghai 200233, China; shaoxiaoping@shsmu.edu.cn; 2Department of Epidemiology, Ministry of Education Key Laboratory of Public Health Safety, Fudan University School of Public Health, Shanghai 200032, China; xylu20@fudan.edu.cn (X.L.); weiyuzhou21@m.fudan.edu.cn (W.Z.); luyihan@fudan.edu.cn (Y.L.)

**Keywords:** Chinese female health care workers, health belief model, human papillomavirus vaccine, vaccination behavior, vaccine preference

## Abstract

Human papillomavirus (HPV) vaccination has been proven to be the most effective method to prevent cervical cancer. This study aimed to determine the HPV vaccination behavior and preference in Chinese female health care workers. A nationwide cross-sectional study was performed to recruit 15,967 respondents aged 18–45 years from 31 provinces in China’s mainland in 2021. Of them, 30.0% have been vaccinated or have made an appointment. Regardless of actual vaccination status, respondents mostly preferred the 9-valent HPV vaccine (58.6%), followed by 4-valent (15.6%) and 2-valent vaccines (3.1%); additionally, 17.9% did not have a preference. Moreover, health beliefs on HPV and HPV vaccination were measured using a health belief model (HBM) analysis. Six HBM constructs differed significantly by HPV vaccination status. Higher levels of perceived susceptibility (beta = 0.074), perceived benefit (beta = 0.072), self-efficacy (beta = 0.304), and cues to action (beta = 0.039) scales were significantly associated with increasing HPV vaccine uptake. In contrast, perceived severity (beta = −0.019) and perceived barriers (beta = −0.089) were negative factors. In conclusion, HPV vaccine uptake is high in Chinese female health care workers. HBM constructs may be effective in facilitating the improvement and delivery of targeted intervention programs to increase HPV vaccine uptake.

## 1. Introduction

Human papillomavirus (HPV) infection is highly prevalent worldwide. Moreover, multiple oncogenic HPV types can cause precancerous cervical lesions and, subsequently, cervical cancer in women [1]. It is estimated that a total of 342,000 deaths worldwide from cervical cancer occurred in 2020, and 428,000 deaths will occur in 2035 [2]. In China, there are an estimated 110,900 new cases and 59,000 deaths from cervical cancer, according to the national cancer statistics by the National Cancer Center in 2020 [3]. Currently, HPV vaccination is considered an effective measure to avert precancerous cervical lesions and cervical cancer in women, according to the Global Alliance for Vaccines and Immunization (GAVI) and World Health Organization (WHO) [4,5,6]. However, acceptance and uptake of HPV vaccines have been influenced by multiple factors and differ widely across the world. So far, various psychological theories and models have been proposed to help explain human behavior related to health, including vaccination behavior. The most well-known and often-used ones are the health belief model (HBM) [7], the protection motivation theory (PMT) [8], and the theory of planned behavior (TPB) [9,10]. The HBM is a conceptual framework that facilitates explaining and predicting health-promoting behaviors in terms of certain health beliefs [11]. Generally, the HBM consists of concepts as follows: perceived susceptibility, perceived severity, perceived benefits, perceived barriers, cues to action, and self-efficacy. The HBM has previously been used to evaluate beliefs and attitudes toward flu vaccination and hepatitis B vaccination [12,13], as well as the relationship between perceptions and COVID-19 vaccination [14]. It has also been adopted in several studies to explain and predict the intent and behavior of HPV vaccination, such as in female college students in Greece and Thailand, indicating perceived benefits had a positive impact on the HPV vaccination [15,16]. A previous study in Hong Kong found that perceived susceptibility, perceived severity, perceived benefit, perceived self-efficacy, and behavioral intention related to HPV vaccination significantly predicted HPV vaccine uptake [17].

Currently, three imported HPV vaccines (Cervarix^®^, Gardasil^®^, and Gardasil^®^) and two Chinese-made bivalent HPV vaccines (Cecolin^®^ and Walrinvax^®^) have become available in China’s mainland [18]. In China, multiple studies have been conducted to characterize HPV vaccination behavior [19]. Of them, the majority targeted female college students and parents of middle and high school girls, while fewer studies targeted respondents aged 18–45 years [20,21]. Due to higher education levels, monthly income, and professional knowledge, female health workers’ vaccination behavior may differ from other sociodemographic groups. Moreover, it has been documented that the health beliefs of health care workers may influence their vaccination behavior and further impact the general population, such as COVID-19 vaccination and flu vaccination [22]. Thus, our study used the HBM to fully determine the impact of social and psychological factors on HPV vaccination behavior. In addition, this study explored the preference for HPV vaccines in Chinese female health care workers. The findings in this study would facilitate organizing future advocacy campaigns to promote the HPV vaccination.

## 2. Materials and Methods

### 2.1. Study Design

This was a nationwide cross-sectional study based on electronic survey data that were collected from 15 November to 23 December 2021. We included the target population as follows: (1) female health care workers in China’s mainland; (2) respondents aged 18–45 years, as the maximum age is 45 for the licensed HPV vaccines in China’s mainland; and (3) respondents were aware of the study purpose and voluntarily agreed to take part. In order to estimate the proportion of HPV vaccine uptake, for a 95% confidence level with an expected true proportion of 2.8% and 0.28% margin of error [19], a sample size of 13,336 was obtained using the formula:N = (Z_α/2_^2^ × S^2^)/d^2^, α = 0.05, Z_α/2_ = 1.96; S^2^ = P(1 − P)

N = sample size, Z = Z value, p = population variance, and d = margin of error.

The questionnaire was designed based on the HBM as previously described [23] and then developed by a panel of experts in the field, including a statistician, a behavioral psychologist, an epidemiologist, and a clinician. Specifically, the experts edited the questionnaire and validated the content in terms of the fit between each statement of questions and corresponding theoretical variables. By using an electronic survey tool supported by www.wjx.com, a QR code was prepared to link the electronic questionnaire and then distributed to the female health care workers across the country through the networks of nursing associations using a convenience sampling strategy. Participants used their smartphones or computers to access and complete the questionnaire.

This study was approved by the Institutional Review Board of the Fudan University School of Public Health (IRB 00002408 and FWA 00002399) under IRB #2021-09-0919.

### 2.2. Measurements

The survey consisted of questions that assessed (1) sociodemographic background; (2) health beliefs on HPV and HPV vaccination; (3) HPV vaccination behavior; and (4) HPV vaccine preference.

Sociodemographic background: Demographics, such as age, ethnicity, educational level, monthly income, marital status, region of residence, and occupational information (professional title and level of hospitals), were collected.

Health beliefs on HPV and HPV vaccination: HBM-derived items were used to measure the respondents’ perception of HPV and HPV vaccination. Six essential dimensions of health beliefs were measured as follows: (1) perceived susceptibility to HPV in the future (five items; e.g., “I was vulnerable to infection with HPV”); (2) perceived severity of HPV infection (five items; e.g., “It would be very harmful for me if I got HPV”); (3) perceived benefits of HPV vaccination (five items; e.g., “HPV vaccination can protect me from infection”); (4) perceived barriers to HPV vaccination (three items; e.g., “The HPV vaccines might have side effects”); (5) self-efficacy for HPV vaccination (three items; e.g., “I believe I can deal with side effects of the HPV vaccines with doctors’ help”); and (6) cues to action refer to external recommendations that might affect individuals’ health-related behaviors. Participants’ responses were divided into 5 agree-disagree scales (1 = strongly disagree, 2 = disagree, 3 = neutral, 4 = agree, 5 = strongly agree).

HPV vaccination behavior and vaccine preference: HPV vaccination status was measured using a one-item question (Have you made an HPV vaccination appointment or been vaccinated against HPV?) with four options (“have been vaccinated”, “have made an appointment but have not been vaccinated yet”, “have an intent to receive vaccination but have not made an appointment yet”, and “have no intent to receive vaccination”). We collapsed the first and second options together and defined it as “Have been vaccinated or have made an appointment”. Moreover, respondents were asked to choose their preferred HPV vaccine, regardless of actual vaccination status or actual vaccines they have been vaccinated or could access, including 9-valent HPV vaccine (9vHPV), 4-valent HPV vaccine (4vHPV), 2-valent HPV vaccine (2vHPV), no preference, and not intending to receive a vaccination.

### 2.3. Statistical Analysis

Descriptive data were presented as mean ± standard deviation (SD) for continuous variables and frequency/percentage for categorical variables. We determined the factors associated with vaccination behavior using multivariable logistic regression; the adjusted odds ratios (OR) were used as the point estimates. Health beliefs on HPV were compared by HPV vaccination status using one-way ANOVA. The adoption of the ANOVA has been verified, including normality (Kolmogorov–Smirnov test, *p* = 0.195), homogeneous variance (*p* = 0.163), independence between the groups of respondents, and absence of outliers. Subsequently, a structural equation model was established to assess the correlation between HBM constructs and vaccination behavior. The standardized estimates of pathway coefficients and the correlation between the related constructs were provided. The beta (β) value represents the effect of the structural equation model. The goodness of fit was evaluated by the goodness of fit index (GFI), adjusted goodness of fit index (AGFI), comparative fitting index (CFI), incremental fitting index (IFI), Tacker–Lewis index (TLI), root mean square error of approximation (RMSEA), and standardized root mean squared residual (SRMR). The criteria for the goodness of fit were GFI > 0.90, AGFI > 0.90, CFI > 0.90, IFI > 0.90, TLI > 0.90, RMSEA < 0.08, and SRMR < 0.08. The IBM SPSS Statistics 23.0 and Amos software version 24.0 (Armonk, NY, USA) were utilized to perform the analysis. All tests were two-sided, with a *p*-value less than 0.05 considered statistically significant.

## 3. Results

### 3.1. Characteristics of Respondents

A total of 15,967 female health care workers aged 18–45 years completed the questionnaires. This study received responses from 31 provinces in China’s mainland, including 10,779 (67.5%) respondents in eastern China, 1232 (7.7%) in northeastern China, 1103 (6.9%) in northwestern China, 961 (6.0%) in central China, 955 (5.9%) in northern China, 486 (3.1%) in southwestern China, and 451 (2.9%) in southern China (Table 1), covering 55 out of 86 prefectural-level districts in four municipalities and 279 out of the 293 prefectural-level cities in China. The mean age of the respondents was 30.6 ± 6.2 years, with 4821 (30.2%) aged 18–26 years, 7713 (48.3%) aged 27–35 years, and 3433 (21.5%) aged 36–45 years. The majority of the respondents were married (61.1%), Han Chinese (93.3%), had a bachelor’s degree (68.3%), primary professional title (57.7%), worked in tertiary hospitals (74.2%), and had monthly income less than CNY 8000 (approximately USD 1200) (82.3%) (Table 1).

### 3.2. Factors Associated with HPV Vaccination Behavior

Among the respondents, 4783 (30.0%) had been vaccinated or had made an appointment. They were more likely to have been vaccinated or made an appointment when they aged 18–26 years (OR = 8.01, 95% CI: 5.95, 10.78) or 27–35 years (OR = 2.54, 95% CI: 2.13, 3.04), worked in the primary and secondary hospitals (OR = 1.35, 95% CI: 1.15, 1.57), and were child-free (OR = 1.23, 95% CI: 1.01, 1.48) (Table 1). In contrast, respondents who had a bachelor degree (OR = 0.81, 95% CI: 0.72, 0.97) and an associate’s degree or below (OR = 0.57, 95% CI: 0.39, 0.84), were married (OR = 0.72, 95% CI: 0.63, 0.81), had a monthly income between CNY 4001 and 8000 (OR = 0.94, 95% CI: 0.89, 0.99), and less than or equal to CNY 4000 (OR = 0.71, 95% CI: 0.63, 0.79) were more unlikely to intend to receive HPV vaccination (Table 1).

Moreover, respondents aged 18–26 years (OR = 5.43, 95% CI: 4.10, 7.20) or 27–35 years (OR = 2.70, 95% CI: 2.29, 3.17) were more likely to have the intent to receive the vaccination but had not made an appointment. In contrast, those who worked in the primary and secondary hospitals (OR = 0.84, 95% CI: 0.72, 0.97), had a monthly income between CNY 4001 and 8000 (OR = 0.88, 95% CI: 0.71, 0.97), and less than or equal to CNY 4000 (OR = 0.92, 95% CI: 0.76, 0.98), were more unlikely to intend to receive HPV vaccination (Table 1).

In addition, vaccination status varied widely among the respondents from different regions of residence, with those in southwestern and eastern China having higher percentages (35.6% and 32.9%, respectively) of having been vaccinated or having made an appointment (Table 1).

### 3.3. HPV Vaccine Preference

Among the respondents who had been vaccinated or made an appointment, 2073 (43.3%) actually chose 9vHPV, 2041 (42.7%) chose 4vHPV, and 669 (14.0%) chose 2vHPV. When they were asked to choose the HPV vaccine they mostly preferred, 75.1% preferred 9vHPV, followed by 4vHPV (15.4%) and 2vHPV (3.2%). Overall, among all the respondents, 9355 (58.6%) preferred 9vHPV, 2494 (15.6%) preferred 4vHPV, 495 (3.1%) preferred 2vHPV, 2861 (17.9%) did not have a preference, and 762 (4.8%) had no intent to receive vaccination, regardless of actual vaccination status (Table 2).

Moreover, vaccination preference was negatively associated with age (*p* < 0.001). Respondents aged 18–26 years had the highest intent to receive HPV vaccination, of which the vast majority (85.4%) preferred 9vHPV (Table 2). Furthermore, respondents who were unmarried (non-divorced and non-widow), with no or primary professional titles, with low monthly income, and with no child preferred 9vHPV in a high proportion (>60%), which might be consistent with the changing trend in age (Table 2); however, their preferences for 4vHPV and 2vHPV and no preference varied disproportionally across the groups.

Interestingly, a considerable percentage of those more than 26 years (47.0%) remained preferred 9vHPV, though they were overage for 9vHPV vaccination according to the license in China. In addition, they shared very similar preferences for 4vHPV (21%) and 2vHPV (4%) between the groups 27–35 years and 36–45 years.

### 3.4. Health Beliefs on the HPV Vaccination

The health beliefs of the respondents on the HPV vaccination comprised the mean scales and related subscale ratings. The internal validity (reliability) of the HBM items in each scale was measured using Cronbach’s alpha. All the Cronbach’s alpha values were above 0.6 and hence were accepted (Table 3). The respondents were divided into three groups based on HPV vaccination behavior. Mean scores of the total perceived susceptibility, perceived severity, perceived benefit, self-efficacy, and cues to action were significantly higher in the group of those who had been vaccinated or had made an appointment (*p* < 0.001) (Table 3). In contrast, those who had not made an appointment or did not intend to receive it had a higher mean score for perceived barriers.

The effects of the model of HBM constructs on vaccination behavior are presented in Figure 1. This model was identified given the good fit indices (GFI = 0.915, AGFI = 0.955, CFI = 0.960, IFI = 0.960, TLI = 0.916, RMSEA = 0.076, SRMR = 0.026) for all the samples. Standardized direct, indirect, and total effects of paths on cues to action, self-efficacy, and vaccination behavior were presented (Table 4). It indicated that four perceived constructs directly affected cues to action, in which perceived susceptibility (beta = 0.080) and perceived benefit (beta = 0.173) positively affected cues to action, while perceived severity (beta = −0.070) and perceived barriers (beta = −0.045) had a negative impact.

Moreover, all six constructs had a direct or indirect impact on vaccination behavior (Table 4, Figure 1). Higher perceived susceptibility, perceived benefit, self-efficacy, and cues to action were significantly associated with increasing HPV vaccination; self-efficacy was especially the largest predictor of vaccination behavior (beta = 0.304). In contrast, perceived severity (beta = −0.019) and perceived barriers (beta = −0.089) were negative factors.

## 4. Discussion

This study characterized the HPV vaccination behavior in Chinese female health care workers in a nationwide survey. A total of 22.8% of the respondents had received the HPV vaccination, and 7.16% had made an HPV vaccination appointment, which was much higher than 2.8% among females aged 9–45 previously reported in Shanghai in 2017–2019 [19], suggesting a substantial increase in the vaccine uptake in a more professional population. Previous studies mostly focused on female adolescents and general female adults instead of health care workers. A study among medical students in Alabama, the USA, found that 32.1% reported completion of HPV vaccination while 15.2% reported partial completion, slightly higher than that in our study [24]. So far, there have been several studies in China to explore the knowledge of the HPV vaccine and vaccination intent among college students or adolescents’ parents [20]. In contrast, our study provided evidence of HPV vaccination for adult women. Furthermore, the HPV vaccination behavior differed significantly across the country and was associated with HPV and HPV vaccination-related health beliefs as determined by HBM constructs. Data by the National Central Cancer Registry of China (NCCRC) showed that cervical cancer mortality rates were lower in Western China (4.16 per 100,000) and Eastern China (2.79 per 100,000) than in Central China (4.43 per 100,000) [25]. It has raised an urgent public health challenge how to increase the HPV vaccine uptake in the areas with high incidences of cervical cancer.

Moreover, HBM constructs were found to be significantly associated with vaccination behavior in this study. In particular, respondents who perceived susceptibility to HPV, believed in the benefits of vaccination (including prevention of diseases and benefits to life, work, family, and society), had higher self-efficacy, and more cues to action, were more likely to receive the HPV vaccination. In contrast, the perceived severity of HPV infection and barriers induced by the vaccine (such as high cost, inconvenient vaccination, and adverse effects) were negatively associated with their behavior. In previous studies, HBM had been utilized to determine the factors associated with HPV vaccination, including perceived barriers being associated with non-vaccination, whereas perceived susceptibility, perceived severity, perceived benefits, and perceived recommendation were associated with increased vaccination in female college students [17]. Similar HBM outcomes had also been documented in a previous Chinese study, in which increasing perceived severity was correlated with a parental preference for HPV vaccine uptake for their daughters [26]. A study predicted intention to receive the COVID-19 vaccine among Israeli adults found that respondents were more likely to be willing to get vaccinated if they reported higher levels of perceived benefits of the COVID-19 vaccine, of perceived severity of COVID-19 infection and of cues to action, according to HBM [27,28]. In addition, a Greek study used extended HBM to examine the role of beliefs in predicting intent to be vaccinated against COVID-19. This study highlighted the interaction effects among the HBM components, consistent with the findings in our study [29].

This study also found that four HBM constructs (perceived susceptibility, perceived severity, perceived benefit, and perceived barriers) directly affected HPV vaccination behavior or indirectly affected self-efficacy and cues to action. Cues to action were correlated with self-efficacy, while self-efficacy was directly correlated with vaccination behavior. Previous studies found that cues to action could be interpreted as recommendations from family, friends, health care workers, media, and academic lectures, especially cues to action from social relationships, which may have a greater impact on attitudes towards vaccination [14,30]. In our study, the direct effect of cues to action was relatively minor; in contrast, its indirect effect or mediating effect was larger. Thus, our study suggested that, on the one hand, improving knowledge of HPV susceptibility, severity, and benefits is particularly important for increasing vaccine uptake. On the other hand, it may also be achieved by cues to action, which requires health care workers, community vaccination sites, and media to make joint efforts to promote knowledge of HPV and HPV vaccines.

This study determined the preference for HPV vaccines. The 9vHPV remained the most popular HPV vaccine (58.6%) in female health care workers across the country, even in those who had received vaccination or had made an appointment (43.3%) and those who were >26 years and overage (47.0%). Furthermore, intent to receive any HPV vaccine was generally high (>90%) across the sociodemographic groups. Given the evidence of vaccine hesitancy in receiving HPV vaccination in the general population [23], it highly warrants immediate strategies aimed at increasing HPV vaccine uptake among women aged 18–45 years in China. One notable example is the proposed “semi-mandatory HPV vaccination strategy”, in which government subsidizes HPV vaccination targeted at low-income settings for high-risk individuals willing to pay an affordable cost [26]. However, in our study, it was noted that there was less difference in the preference for 9vHPV, 4vHPV, 2vHPV, and no preference among the groups stratified by monthly income, suggesting that vaccine price might not be a principal barrier to the HPV vaccination in female health care workers. Further feasible strategies may be developed and characterized by sociodemographics.

Several limitations of the present study should be considered when interpreting our findings. First, we limited our sample to female health care workers in China who had higher education levels, monthly income, and knowledge, which confined the generalizability of the findings to all the females nationwide. However, we were able to obtain a robust sample throughout the country, corresponding to 100% of the prefectures in eastern China, 100% in central China, 84.6% in southern China, 71.1% in southwestern China, 96.8% in northern China, 100% in northeastern China, and 100% in northwestern China. The extensive geographical coverage and diverse settings may help inform us about the overall situation of HPV vaccination. Second, it may take more than 10 min to complete the electronic questionnaire, which might lead to random responses by some respondents without reflecting the real perception. We checked the records and did not identify evident logical errors. In addition, electronic questionnaires have an inherent bias, such as respondents giving their husbands, sons, and daughters the questionnaire to complete for them. Third, in the study, we distributed the questionnaire through the networks of physicians and nursing associations; the majority of respondents were nurses (73.8%), in addition to physicians (19.6%) and pharmacists (6.6%). The nurses had a little lower percentage of having been vaccinated or having made an appointment (28.5%) compared to the physicians (34.2%) and pharmacists (32.9%). It might provide limited generalizability to diverse professions. Fourth, we did not investigate when respondents had been vaccinated against HPV. This study aimed to determine the vaccination status among female health care workers and then explored the influence of health beliefs on vaccination behavior. Diverse vaccination age may be considered as a result of health beliefs. Additionally, the payment mechanism for HPV vaccines among female adults has not changed since HPV vaccines become available in China’s mainland. Also, they might not have received the HPV vaccines before becoming adults, as less than 5% of female adolescents and children have been vaccinated against HPV in China’s mainland.

## 5. Conclusions

This nationwide study found that Chinese female health care workers had high HPV vaccine uptake. Meanwhile, they frequently preferred multi-valent HPV vaccines, regardless of vaccination status. Moreover, HPV vaccination behavior was determined to be associated with higher perceived susceptibility and benefit, self-efficacy, cues to action, and lower perceived severity and barriers, when using an HBM analysis. It suggested that the interventions targeting HBM constructs may be effective in increasing the HPV vaccine uptake. Currently, China has changed its policy and countermeasures against COVID-19; consequently, HPV vaccination has gradually increased in China. Thus, this study has important implications in facilitating the delivery of certain intervention programs to enhance HPV vaccination.

## Figures and Tables

**Figure 1 vaccines-11-01367-f001:**
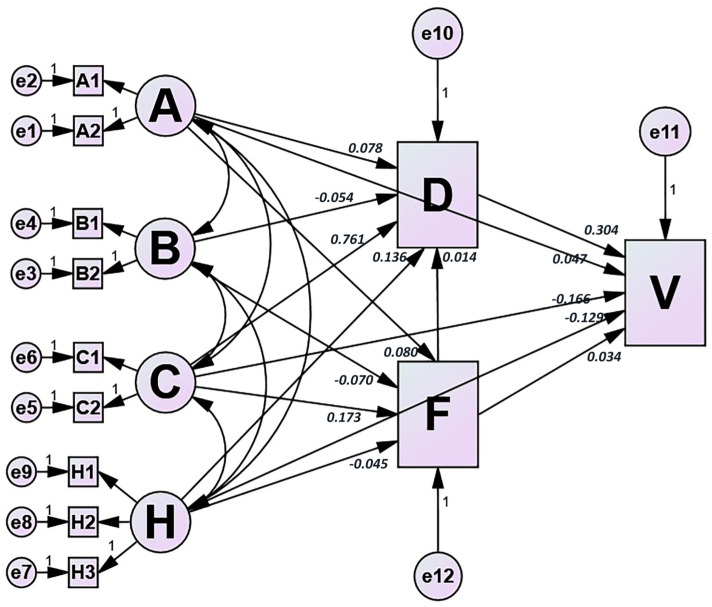
Effects of health belief model (HBM) constructs on the HPV vaccination behavior in Chinese female health care workers. The path coefficients were displayed with a significance of *p* value less than 0.05. The model fit indices were as follows: goodness of fit index (GFI) = 0.915; adjusted goodness of fit index (AGFI) = 0.955; comparative fit index (CFI) = 0.960; incremental fitting index (IFI) = 0.960; Tacker–Lewis index (TLI) = 0.916; root mean square error of approximation (RMSEA) = 0.076; standardized root mean squared residual (SRMR) = 0.026. The HBM constructs were presented as follows: A, perceived susceptibility; B, perceived severity; C, perceived benefit; D, self-efficacy; H, perceived barriers; F, cues to action; V, vaccination behavior.

**Table 1 vaccines-11-01367-t001:** Factors associated with HPV vaccination by multivariable logistic regression.

Sociodemographic Groups	No. Respondents	Have Been Vaccinated or Have Made an Appointment		Have an Intent to Receive Vaccination but Have Not Made an Appointment		Have Not Intended to Receive (Reference Group)
		No. (%)	OR (95% CI)	No. (%)	OR (95% CI)	No. (%)
Age (years)						
18–26	4821	1825 (37.9)	8.01 (5.95, 10.78) *	2847 (59.1)	5.43 (4.10, 7.20) *	149 (3.1)
27–35	7713	2045 (26.5)	2.54 (2.13, 3.04) *	5188 (67.3)	2.70 (2.29, 3.17) *	480 (6.2)
36–45	3433	913 (26.6)	1.00	2030 (59.1)	1.00	490 (14.3)
Ethnicity						
Han Chinese	14,912	4565 (30.6)	1.35 (0.99, 1.83)	9313 (62.5)	1.25 (0.95, 1.65)	1034 (6.9)
Minority Chinese	1055	218 (20.7)	1.00	752 (71.3)	1.00	85 (8.1)
Educational level						
Associate’s degree or below	4572	1351 (29.5)	0.57 (0.39, 0.84) *	2927 (64.0)	1.08 (0.74, 1.56)	294 (6.4)
Bachelor degree	10,906	3254 (29.8)	0.81 (0.72, 0.97) *	6867 (63.0)	1.26 (0.89, 1.78)	785 (7.2)
Master’s degree or above	489	178 (36.4)	1.00	271 (55.4)	1.00	40 (8.2)
Marital status						
Married	9763	2549 (26.1)	0.72 (0.63, 0.81) *	6372 (65.3)	1.09 (0.86, 1.39)	842 (8.6)
Unmarried	6204	2234 (36.0)	1.00	3693 (59.5)	1.00	277 (4.5)
Professional title						
No title	1470	424 (28.8)	0.61 (0.43, 0.86) *	983 (66.9)	0.90 (0.65, 1.25)	63 (4.3)
Primary professional title	9220	2895 (31.4)	0.84 (0.70, 0.98) *	5783 (62.7)	0.86 (0.73, 1.02)	542 (5.9)
Middle professional title or above	5277	1464 (27.7)	1.00	3299 (62.5)	1.00	514 (9.7)
Hospital level						
Primary and secondary hospital	4126	1528 (37.0)	1.35 (1.15, 1.57) *	2286 (55.4)	0.84 (0.72, 0.97) *	312 (7.6)
Tertiary hospital	11,841	3255 (27.5)	1.00	7779 (65.7)	1.00	807 (6.8)
Monthly income (CNY)						
≤4000	7327	2044 (27.9)	0.71 (0.63, 0.79) *	4830 (65.9)	0.88 (0.71, 0.97) *	453 (6.2)
4001–8000	5807	1833 (31.5)	0.94 (0.89, 0.99) *	3547 (61.1)	0.92 (0.76, 0.98) *	407 (7.0)
≥8001	2833	906 (32.0)	1.00	1688 (59.6)	1.00	259 (9.1)
Region of residence in China						
Southwestern	486	173 (35.6)	4.01 (2.35, 6.82) *	291 (59.9)	1.29 (0.78, 2.12)	22 (4.5)
Eastern	10,779	3550 (32.9)	2.35 (1.72, 3.21) *	6434 (59.7)	0.86 (0.66, 1.13)	795 (7.4)
Central	961	290 (30.2)	3.12 (2.00, 4.87) *	630 (65.6)	1.36 (0.91, 2.04)	41 (4.3)
Southern	451	123 (27.3)	6.44 (3.19, 13.00) *	318 (70.5)	3.21 (1.64, 6.28) *	10 (2.2)
Northeastern	1232	304 (24.7)	1.73 (1.20, 2.51) *	830 (67.4)	0.90 (0.65, 1.25)	98 (8.0)
Northern	955	196 (20.5)	1.71 (1.23, 2.60) *	699 (73.2)	1.08 (0.74, 1.56)	60 (6.3)
Northwestern	1103	147 (13.3)	1.00	863 (78.2)	1.00	93 (8.4)
Child						
No child	7199	2548 (35.4)	1.23 (1.01, 1.48) *	4327 (60.1)	0.98 (0.77, 1.24)	324 (4.5)
Have child	8768	2235 (25.5)	1.00	5738 (65.4)	1.00	795 (9.1)

* each *p* < 0.05.

**Table 2 vaccines-11-01367-t002:** Preferences for HPV vaccines across demographic characteristics.

Sociodemographic Groups	9-Valent HPV Vaccine	4-Valent HPV Vaccine	2-Valent HPV Vaccine	No Preference	Not Inend to Receive Vaccination	*p* Value
No. respondents	9355	2494	495	2861	762	
Age (years)						
18–26	4115 (85.4)	128 (2.7)	54 (1.1)	436 (9.0)	88 (1.8)	<0.001
27–35	3812 (49.4)	1637 (21.2)	304 (3.9)	1628 (21.1)	332 (4.3)	
36–45	1428 (41.6)	729 (21.2)	137 (4.0)	797 (23.2)	342 (10.0)	
Ethnicity						
Han Chinese	8898 (59.7)	2332 (15.6)	385 (2.6)	2605 (17.5)	692 (4.6)	<0.001
Minority Chinese	457 (43.3)	162 (15.4)	110 (10.4)	256 (24.3)	70 (6.6)	
Educational level						
Associate’s degree or below	3051 (66.7)	433 (9.5)	131 (2.9)	753 (16.5)	204 (4.5)	<0.001
Bachelor degree	6026 (55.3)	1973 (18.1)	350 (3.2)	2029 (18.6)	528 (4.8)	
Master’s degree or above	278 (56.9)	88 (18.0)	14 (2.9)	79 (16.2)	30 (6.1)	
Marital status						
Married	4600 (47.1)	2067 (21.2)	401 (4.1)	2105 (21.6)	590 (6.0)	<0.001
Unmarried	4755 (76.6)	427 (6.9)	94 (1.5)	756 (12.2)	172 (2.8)	
Unmarried without partner (n = 3776)	2965 (78.5)	202 (5.3)	45 (1.2)	463 (12.3)	101 (2.7)	
Unmarried with partner (n = 2142)	1680 (78.4)	168 (7.8)	34 (1.6)	216 (10.1)	44 (2.1)	
Divorced and widow (n = 286)	110 (38.5)	57 (19.9)	15 (5.2)	77 (26.9)	27 (9.4)	
Professional title						
No title	1165 (79.3)	60 (4.1)	20 (1.4)	188 (12.8)	37 (2.5)	<0.001
Primary professional title	5830 (63.2)	1225 (13.3)	279 (3.0)	1509 (16.4)	377 (4.1)	
Middle professional title or above	2360 (44.7)	1209 (22.9)	196 (3.7)	1164 (22.1)	348 (6.6)	
Hospital level						
Primary and secondary hospital	2332 (56.5)	711 (17.2)	144 (3.5)	722 (17.5)	217 (5.3)	0.001
Tertiary hospital	7023 (59.3)	1783 (15.1)	351 (3.0)	2139 (18.1)	545 (4.6)	
Monthly income (CNY)						
≤4000	4500 (61.4)	1021 (13.9)	185 (2.5)	1319 (18.0)	302 (4.1)	<0.001
4001–8000	3326 (57.3)	954 (16.4)	235 (4.0)	1016 (17.5)	276 (4.8)	
≥8001	1529 (54.0)	519 (18.3)	75 (2.6)	526 (18.6)	184 (6.5)	
Region of residence in China						
Northern	505 (52.9)	175 (18.3)	23 (2.4)	209 (21.9)	43 (4.5)	<0.001
Northeastern	668 (54.2)	198 (16.1)	69 (5.6)	239 (19.4)	58 (4.7)	
Northwestern	473 (42.9)	145 (13.1)	105 (9.5)	303 (27.5)	77 (7.0)	
Central	582 (60.6)	146 (15.2)	21 (2.2)	186 (19.4)	26 (2.7)	
Eastern	6529 (60.6)	1682 (15.6)	255 (2.4)	1776 (16.5)	537 (5.0)	
Southern	276 (61.2)	73 (16.2)	10 (2.2)	84 (18.6)	8 (1.8)	
Southwestern	322 (66.3)	75 (15.4)	12 (2.5)	64 (13.2)	13 (2.7)	
Child						
No child	5429 (75.4)	549 (7.6)	106 (1.5)	905 (12.6)	210 (2.9)	<0.001
Have child	3926 (44.8)	1945 (22.2)	389 (4.4)	1956 (22.3)	552 (6.3)	

**Table 3 vaccines-11-01367-t003:** Average score (SD) of health beliefs on HPV and HPV vaccination.

	All the Respondents	Have Been Vaccinated or Have Made an Appointment	Have an Intent to Receive Vaccination but Have Not Made an Appointment	Have Not Intended to Receive	*p* Value
Perceived susceptibility					
Lifetime risk of infection (3 items)	11.45 (2.67)	11.71 (2.59)	11.47 (2.64)	10.14 (2.89)	<0.001
Personal risk of infection (2 items)	5.79 (2.03)	5.79 (2.07)	5.84 (2.00)	5.32 (2.01)	0.871
Total (Cronbach ɑ = 0.873)	17.23 (4.09)	17.50 (4.01)	17.12 (4.12)	15.98 (4.04)	<0.001
Perceived severity					
Severity of infection (3 items)	12.04 (2.55)	12.22 (2.46)	12.09 (2.53)	10.88 (2.89)	<0.001
Negative impacts of infection (2 items)	8.35 (1.61)	8.41 (1.57)	8.32 (1.63)	7.75 (1.89)	<0.001
Total (Cronbach ɑ = 0.907)	20.39 (3.82)	20.63 (3.67)	20.29 (3.89)	18.21 (3.76)	<0.001
Perceived benefit					
Prevent diseases (3 items)	12.26 (2.32)	12.44 (2.20)	12.31 (2.29)	10.99 (2.69)	<0.001
Benefits to daily life (2 items)	8.20 (1.62)	8.31 (1.55)	8.26 (1.58)	7.24 (1.87)	<0.001
Total (Cronbach ɑ = 0.950)	20.46 (3.77)	20.75 (3.57)	20.34 (3.85)	18.34 (3.73)	<0.001
Self-efficacy for health-protective behavior					
Total (3 items, Cronbach ɑ = 0.850)	11.97 (2.15)	12.39 (1.99)	11.96 (2.09)	10.34 (2.47)	<0.001
Perceived barriers					
Total (3 items, Cronbach ɑ = 0.763)	10.31 (2.32)	10.12 (2.37)	10.39 (2.29)	11.67 (2.64)	<0.001
Cues to action					
Total (3 items, Cronbach ɑ = 0.811)	9.70 (4.05)	10.04 (4.06)	9.55 (4.05)	9.31 (4.01)	<0.001

**Table 4 vaccines-11-01367-t004:** Total, direct, and indirect effects of health belief model constructs on vaccination behavior.

	Direct	Indirect	Total
Cues to action ← Perceived susceptibility	0.080	0.000	0.080
Cues to action ← Perceived severity	−0.070	0.000	−0.070
Cues to action ← Perceived benefit	0.173	0.000	0.173
Cues to action ← Perceived barriers	−0.045	0.000	−0.045
Self-efficacy ← Perceived susceptibility	0.078	0.001	0.079
Self-efficacy ← Perceived severity	−0.054	−0.001	−0.055
Self-efficacy ← Perceived benefit	0.761	0.002	0.763
Self-efficacy ← Perceived barriers	0.136	−0.001	0.135
Self-efficacy ← Cues to action	0.014	0.000	0.014
Vaccination behavior ← Perceived susceptibility	0.047	0.027	0.074
Vaccination behavior ← Perceived severity	0.000	−0.019	−0.019
Vaccination behavior ← Perceived benefit	−0.166	0.238	0.072
Vaccination behavior ← Self-efficacy	0.304	0.000	0.304
Vaccination behavior ← Perceived barriers	−0.129	0.040	−0.089
Vaccination behavior ← Cues to action	0.034	0.005	0.039

## Data Availability

The datasets generated during and/or analyzed during the current study are not publicly available due to the privacy policy but are available from the corresponding author upon reasonable request.

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
