# Peer review of "HPV Vaccination Behavior, Vaccine Preference, and Health Beliefs in Chinese Female Health Care Workers: A Nationwide Cross-Sectional Study"

_vaccines, 2023, doi:10.3390/vaccines11081367_

Round 1
Reviewer 1 Report
This could have been a very informative study if the reasons behind each preference (for example the different vaccine options) were investigated further. This could be done by conducting interviews of representative participants from different regions of different age groups.
The conclusion does not match the results. For example, Lines 25-26 “ In conclusion, HPV vaccine uptake is high in Chinese female health care workers.” Do you mean that 30% vaccine uptake is high? Which data(s) do you compare with?
The introduction section needs to be made more reader-friendly. All the components (the current situation, the knowledge gap, the methodology chosen for the study, and the rationale for this study etc.) were present but not organized well.
Lines 117-120: Why can’t the four options for “HPV vaccination behavior and vaccine preference” be analyzed separately? The first three options seem more relevant.
A lot of data was generated from the study; However, the interpretation of this data was weak.
For example, Lines 173-174 “southwestern and eastern China having higher percentages (35.6% and 32.9%, respectively) of having been vaccinated or having made an appointment (Table 1).” Do you have any information to correlate HPV-associated disease and cancer incidence with these two regions?
Lines 279-284 only confirm a common understanding.
The majority of participants were nurses (>70%). Did you compare this population with the other two populations [physicians (19.6%) and pharmacists (6.6%)]? I am wondering whether you can observe any new themes.
In your listed limitations, you mentioned the random responses. Did you use any strategies to minimize this possibility?
Line 213-220 What “Beta=” represents was not mentioned in your statistical analysis.
This paper will benefit from a professional editor's help.
Author Response
Reviewer 1
- This could have been a very informative study if the reasons behind each preference (for example the different vaccine options) were investigated further. This could be done by conducting interviews of representative participants from different regions of different age groups.
Authors’ response: Thank you for your valuable comments. This study aims to quantitatively determine the HPV vaccination in Chinese female health care workers, including vaccination behavior, vaccine preference, and their associations with health beliefs. Qualitative study may be further performed to explore the reasons how to further improve the HPV vaccination in Chinese females.
- The conclusion does not match the results. For example, Lines 25-26 “In conclusion, HPV vaccine uptake is high in Chinese female health care workers.” Do you mean that 30% vaccine uptake is high? Which data(s) do you compare with?
Authors’ response: Thank you for your detailed comments. In our study, 22.8% of the respondents had received the HPV vaccination and 7.16% had made HPV vaccination appointment, which was considerably higher compared to 4.3% among females aged 20-45 previously reported in 2017-2019 in Shanghai that is the most socioeconomically developed metropolis in China’s mainland. We have added it in the Discussion.
- The introduction section needs to be made more reader-friendly. All the components (the current situation, the knowledge gap, the methodology chosen for the study, and the rationale for this study etc.) were present but not organized well.
Authors’ response: We appreciate your detailed comments. We have accordingly revised in the Introduction.
- Lines 117-120: Why can’t the four options for “HPV vaccination behavior and vaccine preference” be analyzed separately? The first three options seem more relevant. A lot of data was generated from the study; However, the interpretation of this data was weak.
Authors’ response: Thank you for your suggestion. HPV vaccine supply is usually insufficient across the regions in China’s mainland, especially before the two Chinese-made bivalent HPV vaccines onto the market (2020 and 2023), which leads to Chinese females queuing up for HPV vaccination after making an appointment. If they do not make the appointment, they would not be on the waiting list. In our study, the respondents who chose the first and second options only had difference in vaccination time due to insufficient vaccine supply, rather than difference in vaccination behavior. Thus, we collapsed the first and second options as the respondents that “have made an appointment but have not been vaccinated yet” could be considered identical to those had been vaccinated. Moreover, we agree with you that the third and fourth options should be analyzed separately. We have accordingly revised in the manuscript.
- For example, Lines 173-174 “southwestern and eastern China having higher percentages (35.6% and 32.9%, respectively) of having been vaccinated or having made an appointment (Table 1).” Do you have any information to correlate HPV-associated disease and cancer incidence with these two regions?
Authors’ response: We appreciate your detailed comments. The data on cervical cancer mortality in the female population from 2005 to 2015 in the “China Cancer Registry Annual Report” showed that cervical cancer mortality in western China (4.16 deaths per 100,000) and in eastern China (2.79 deaths per 100,000) were lower than that in central China (4.43 deaths per 100,000). We have added it in the Discussion.
- Lines 279-284 only confirm a common understanding. The majority of participants were nurses (>70%). Did you compare this population with the other two populations [physicians (19.6%) and pharmacists (6.6%)]? I am wondering whether you can observe any new themes.
Authors’ response: Thank you for your detailed comments. In our study, we recruited much fewer physicians (19.6%) and pharmacists (6.6%), compared to nurses (73.8%), so we made only a general conclusion to female health care workers. We have added the comparison in the study limitations.
- In your listed limitations, you mentioned the random responses. Did you use any strategies to minimize this possibility? Line 213-220 What “Beta=” represents was not mentioned in your statistical analysis.
Authors’ response: Thank you for your comments. In the study, we appropriately increased the sample size and double-checked the questionnaire data to minimize the possibility of random responses. Additionally, the beta (β) value is the coefficient representing the effect in the structural equation model. We have added it in the Materials and Methods.
Reviewer 2 Report
This study explored the HPV vaccination status, vaccine preference, and associated factors using the Health Belief Model among Chinese female health care workers . Some issues need to be addressed before considering for publication.
General issues:
- The Health Belief Model was sometimes written as ‘health belief model’ or ‘Health Belief Model’, and ‘mainland China’ or ‘China’s mainland’. Please express them consistently.
Introduction
- A few theories commonly used to explain health-related behaviors were mentioned. However, the authors did not give reasons why HBM was chosen in this study.
- Any statistics about how HPV affects women’s health in China?
- Please talk about why it is important study the populations of female healthcare workers.
- Please talk more about previous studies on how beliefs affected health care workers’ vaccination behaviors mentioned in the last paragraph in the introduction.
- Some studies revealed associations between HPV vaccination and HBM variables in Chinese population. One example is listed below and the authors may cite it when appropriate.
Wang, Z.; Fang, Y.; Chan, P.S.-f.; Chidgey, A.; Fong, F.; Ip, M.; Lau, J.T.F. Effectiveness of a Community-Based Organization—Private Clinic Service Model in Promoting Human Papillomavirus Vaccination among Chinese Men Who Have Sex with Men. Vaccines 2021, 9, 1218. https://doi.org/10.3390/vaccines9111218
Method
- Please give the inclusion and exclusion criteria of the participants.
- Please give the reliability of the measurements if applicable.
Results
- Please use three-line tables which is the most commonly used style for presenting tables in academic paper.
Discussion
- Any studies about uptake of HPV vaccination in female health care workers in other countries? Is the uptake in this study higher or lower than those studies?
-
- Please talk about the practical implications of the study results.
Look forward to reading the revised version of this paper.
Author Response
Reviewer 2
This study explored the HPV vaccination status, vaccine preference, and associated factors using the Health Belief Model among Chinese female health care workers. Some issues need to be addressed before considering for publication.
- General issues: The Health Belief Model was sometimes written as ‘health belief model’ or ‘Health Belief Model’, and ‘mainland China’ or ‘China’s mainland’. Please express them consistently.
Authors’ response: Thank you for your detailed comments. We have revised accordingly in the manuscript.
- Introduction: A few theories commonly used to explain health-related behaviors were mentioned. However, the authors did not give reasons why HBM was chosen in this study.
Authors’ response: Thank you for your insight. Our study used the health belief model to fully determine the impact of social and psychological factors associated with HPV vaccination, as it may be more appropriate to predict the HPV vaccination uptake. We have revised accordingly in the manuscript.
- Any statistics about how HPV affects women’s health in China?
Authors’ response: We appreciate your detailed comments. In China, there are an estimated 110,900 new cases and 59,000 deaths from cervical cancer according to the national cancer statistics by the National Cancer Center in 2020. We have added it in the Introduction.
- Please talk about why it is important study the populations of female healthcare workers.
Authors’ response: Thank you for your comments. In China, multiple studies have been conducted among female college students and parents of middle and high school girls to characterize the HPV vaccination behavior. Due to higher education level, monthly income, and professional knowledge, female health workers’ vaccination behavior may differ from other sociodemographic groups. In addition, several studies have found that health beliefs of health care workers may influence their vaccination behavior and further impact general population, such as COVID-19 vaccination and flu vaccination. We have added it in the Introduction.
- Please talk more about previous studies on how beliefs affected health care workers’ vaccination behaviors mentioned in the last paragraph in the introduction. Some studies revealed associations between HPV vaccination and HBM variables in Chinese population. One example is listed below and the authors may cite it when appropriate.
Wang, Z.; Fang, Y.; Chan, P.S.-f.; Chidgey, A.; Fong, F.; Ip, M.; Lau, J.T.F. Effectiveness of a Community-Based Organization—Private Clinic Service Model in Promoting Human Papillomavirus Vaccination among Chinese Men Who Have Sex with Men. Vaccines 2021, 9, 1218. https://doi.org/10.3390/vaccines9111218
Authors’ response: Thank you for your valuable suggestion. We have added more about previous studies on how health beliefs affected health care workers’ vaccination behavior in the Introduction, including the above mentioned publication.
- Method: Please give the inclusion and exclusion criteria of the participants. Please give the reliability of the measurements if applicable.
Authors’ response: Thank you for your comments. We presented the inclusion criteria in the study design as follows: 1) female health care workers in China’s mainland; 2) respondents aged 18-45 years, as the maximum age is 45 for the licensed HPV vaccines in China’s mainland; and 3) respondents were aware of the study purpose and voluntarily agreed to take part.
The internal validity (reliability) of the HBM items in each scale was measured using Cronbach’s alpha. All the Cronbach’s alpha values were above 0.6 and hence were accepted. The adoption of the ANOVA has been verified, including normality (Kolmogorov-Smirnov test, P=0.195), homogeneous variance (P=0.163), independence between the groups of respondents, and absence of outliers. The health belief model was identified given the good fit indices (GFI = 0.915, AGFI =0.955, CFI= 0.960, IFI =0.960, TLI =0.916, RMSEA = 0.076, SRMR = 0.026) for all the samples.
- Results: Please use three-line tables which is the most commonly used style for presenting tables in academic paper.
Authors’ response: Thank you for your detailed comments. We have revised accordingly in the manuscript.
- Discussion: Any studies about uptake of HPV vaccination in female health care workers in other countries? Is the uptake in this study higher or lower than those studies? Please talk about the practical implications of the study results. Look forward to reading the revised version of this paper.
Authors’ response: Thank you for your valuable comments. Previous studies in other countries mostly focused on the adolescents’ uptake of HPV vaccines. A study among medical students in Alabama, the USA, found that 32.1% reported completion of HPV vaccination while 15.2% reported partial completion, slightly higher than that in our study. We have revised accordingly in the Discussion.
Author Response
Reviewer 3
HPV and the vaccine are important public health topics. Understanding vaccine decisions in China has the potential to contribute to the larger public health issue of vaccination. The study appears to be well executed with a large, diverse sample. There are, however, several concerns, some more significant than others.
- First, we need more information on about the dimensions including sample items.
Authors’ response: Thank you for your comments. We have added more information in the Materials and Methods.
- The text might refer to the reader to the table; however, it is not clear why the paper discussed subdimensions since there is no conceptual reason provided and they are not analyzed. I believe the discussion needs to center around the overall dimensions and the items used to measure them. Reliability information on for the dimensions analyzed should be added. Analyses do not include the subdimensions (prevent disease); rather the overall dimension (perceived benefit) so the reliability data presented are not relevant. Minor issue: I don’t believe these are Likert scales, which require a specific methodology. They are simply agree-disagree scales.
Authors’ response: Thank you for your valuable comments. We discussed the subdimensions because they are observed variables in the structural equation model that should be routinely presented, though the analysis focused on the overall dimension in our study. We have removed the reliability information for the subdimensions, leaving only the reliability information for the overall dimensions (Table 3).
Additionally, we have revised accordingly in the Materials and Methods as follows: “Participants’ responses were divided into 5 agree-disagree scales (1= strongly disagree, 2= disagree, 3= neutral, 4= agree, 5= strongly agree).”
- Second, I do not agree with collapsing the vaccinated and plan to vaccinate categories. These can be very different groups. Some people plan to vaccinate but do not initiate and others initiate and do not complete. The analyses should look at these groups separately. HPV vaccination status was measured using a one-item question (Have you made an HPV vaccination appointment or been vaccinated against HPV?) with four options (‘have been vaccinated’, ‘have made an appointment but have not been vaccinated yet’, ‘have an intent to receive vaccination but have not made an appointment yet’, and ‘have no intent to receive vaccination’).
Authors’ response: Thank you for your suggestion. HPV vaccine supply is usually insufficient across the regions in China’s mainland, especially before the two Chinese-made bivalent HPV vaccines onto the market (2020 and 2023), which leads to Chinese females queuing up for HPV vaccination after making an appointment. If they do not make the appointment, they would not be on the waiting list. In our study, the respondents who chose the first and second options only had difference in vaccination time due to insufficient vaccine supply, rather than difference in vaccination behavior. Thus, we collapsed the first and second options as the HPV vaccination as the respondents that “have made an appointment but have not been vaccinated yet” could be considered identical to those had been vaccinated. Moreover, we agree with you that the third and fourth options should be analyzed separately. We have accordingly revised in the manuscript.
- While the study was well-executed and HPV vaccination on is an important topic, like many projects using the HBM the findings are not particularly compelling. Isn’t it obvious is a vaccine is seen as beneficial and you are vulnerable to the underlying disease, you are likely to get vaccinated? If the “cues to action” were more clearly explained the project might have more perceived value. Suggest shortening the paper (e.g., eliminate the demographics table and describe in text) to reflect the contribution. It also would increase the contributions to note evidence-based HPV vaccination promotion programs that exist.
Authors’ response: Thank you for your insight. We agree with you that perceived benefits that would be obtained through the vaccination are observable; however, perceived severity and perceived barriers might be interfere with health promoting behaviors. We prefer to remain the demographics table to facilitate potential readers understanding what respondents had been included in the study. Moreover, we have added more on the interpretation of cues to action within the HBM in the Discussion.
Reviewer 4 Report
Dear authors,
Thank you for giving me the opportunity to review this paper.
A well designed and presented paper. HPV vaccination is a powerful preventive measure, and healthcare workers have an impact to the vaccination coverage of the target population.
Authors comment in "results" section the vaccine-type preference. They could try to give an explanation about the preferences.
Author Response
Reviewer 4
- Dear authors, thank you for giving me the opportunity to review this paper. A well designed and presented paper. HPV vaccination is a powerful preventive measure, and healthcare workers have an impact to the vaccination coverage of the target population. Authors comment in "results" section the vaccine-type preference. They could try to give an explanation about the preferences.
Authors’ response: Thank you for your detail comments. Respondents were asked to choose preferred HPV vaccine, including 9-valent HPV vaccine, 4-valent HPV vaccine, 2-valent HPV vaccine, no preference, and being not intent to receive vaccination, regardless of their HPV vaccination status.
Round 2
Reviewer 1 Report
The authors have addressed all the concerns raised previously. The revised version reads better now.
Reviewer 2 Report
Thank you for addressing my comments.